# A Personality-Driven Recommender System for Cross-Domain Learning Based on Holland Code Assessments

**Ja-Hwung Su** [1] **, Yi-Wen Liao** [2,*] **, Jia-Zhen Xu** [2] **and Yu-Wei Zhao** [2]

[1] Department of Computer Science and Information Engineering, National University of Kaohsiung, Kaohsiung 811, Taiwan; bb0820@ms22.hinet.net
[2] Department of Information Management, Cheng Shiu University, Kaohsiung 833, Taiwan; 40611135@gcloud.csu.edu.tw (J.-Z.X.); 40611136@gcloud.csu.edu.tw (Y.-W.Z.)
[*] Correspondence: k0632@gcloud.csu.edu.tw

**Abstract:** Over the past few decades, AI has been widely used in the field of education. However, very little attention has been paid to the use of AI for enhancing the quality of cross-domain learning. College/university students are often interested in different domains of knowledge but may be unaware of how to choose relevant cross-domain courses. Therefore, this paper presents a personality-driven recommender system that suggests cross-domain courses and related jobs by computing personality similarities and probable course grades. In this study, 710 students from 12 departments in a Taiwanese university conducted Holland code assessments. Based on the assessments, a comprehensive empirical study, including objective and subjective evaluations, was performed. The results reveal that (1) the recommender system shows very promising performances in predicting course grades (objective evaluations), (2) most of the student testers had encountered difficulties in selecting cross-domain courses and needed the further support of a recommender system, and (3) most of the student testers positively rated the proposed system (subjective evaluations). In summary, Holland code assessments are useful for connecting personalities, interests and learning styles, and the proposed system provides helpful information that supports good decision-making when choosing cross-domain courses.

**Keywords:** artificial intelligence; cross-domain learning; recommender system; Holland code; personality

## 1. Introduction

The use of artificial intelligence for educational purposes has been studied for several years. In this field, learning and teaching are the two main processes that attract the most research attention. Whether the subject is learning or teaching, the primary goal is to increase the quality of education. As described in a previous study [1], learning can be divided into three stages, namely, before learning, learning and after learning. Moreover, many past studies have shown that high learning performance depends heavily on learning interest [1–3], and the appropriate career is highly related to the individual's personality [4]. Therefore, effectively associating learning interest with personality is a very important issue to consider before learning. Although much research has investigated the links between learning interest and personality, there are currently no methods that cater to cross-domain learning demands. This is a significant gap because students often have multiple domain interests. For example, a student might be interested in science and business, but she/he may not know which courses are relevant. Figure 1, derived from a previous work [1], illustrates the division of first-stage learning into two types, namely, single-domain learning and cross-domain learning. In single-domain learning, students acquire knowledge from one domain, whereas they learn multiple domains in cross-domain learning.

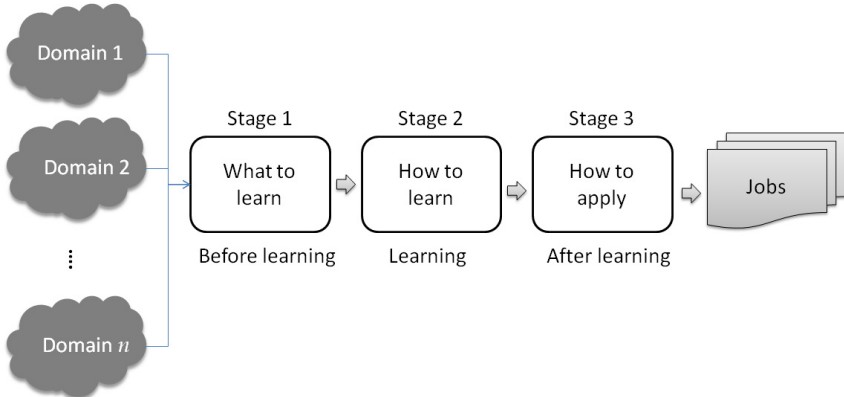

**Figure 1.** Stages of cross-domain learning.

When students need guidance to determine what to learn from multiple domains, a recommender system can serve as an effective solution by providing them with useful information. In principle, a recommender system uses a set of learning algorithms to discover user preferences to yield useful recommendations [5]. Generally, user preferences are represented by two types, namely, explicit and implicit preferences. Explicit preferences are identified from a user's ratings of items with scores of 1–5, where 1 and 2 indicate a negative answer and 3, 4 and 5 indicate an affirmative response. A user's implicit preferences are hidden in their behaviors, such as social media tags, navigation logs, browsing history, etc. Therefore, in this field, determining how to effectively discover the user-to-item affinity based on an individual's preferences has become a challenging issue in recent years. To explore this issue, a number of recent recommender techniques have been proposed, including collaborative filtering, emotional-based, content-based, demographic and knowledge-based recommender systems [6–8]. With these techniques, in the past few years, recommender systems have been adopting an increasingly prominent role because of their multi-domain applicability and the abundance of applications that provide personalized services [9–12]. Thus, recommender systems can be regarded as being due to the implementation of various artificial intelligence methods [13–16].

In the method proposed in this paper, the aim is to provide good recommendations to guide students towards their interest areas in multiple domains in addition to a single domain. The primary innovation of this method over traditional learning systems is three-fold.

- In terms of learning stages, the proposed method focuses on the first stage of "what to learn" instead of "how to learn" and "how to apply". This is because a person's job is highly related to her/his learning direction [17–19], and the learning direction is implied by the learning interest. Therefore, our intent is to leverage the learning interest to determine what the student needs to learn.
- In terms of what to learn, the major difference between this paper and previous works [1,20] is that the proposed system aims to provide useful recommendations when a student faces a number of cross-domain courses. With this information, cross-domain learning achievements can be significantly enhanced.
- In terms of discovering cross-domain interests, in the proposed approach, Holland codes [21] are used as personality features and form the basis on which personality similarities are computed. According to these similarities, the student's potential grades in cross-domain courses of interest are predicted.

To test the proposed method, we developed a recommender system that suggests a set of cross-domain courses and a set of relevant jobs to students. Through this system, objective and subjective evaluations were further conducted. The evaluation results reveal that the proposed system is very effective for determining courses in different domains that the user will potentially find interests.

The rest of this paper is laid out as follows. In Section 2, the related literature is

summarized. After the literature review, in Section 3, the proposed approach is described step by step. The experimental study is then presented in Section 4. In the final section, we draw conclusions and describe future work.

## 2. Previous Studies

In recent years, cross-domain learning has become a trend because many jobs require more than one skill. Therefore, discovering multiple interests for students has been a hot topic in the field of educational science. There are five issues to consider for cross-domain learning, namely, personality, interest, learning, achievement and career. These have been studied by many previous researchers and are summarized in Figure 2. In Figure 2, two main concepts are illustrated: (1) personality is a core factor that is relevant to interest, learning style, learning achievement and career, and (2) a sequence of cause and effect is hidden in these relations, where the personality implies the interest, the interest implies the learning style and career, the learning style further affects learning achievement [22,23], and finally, the career will highly rely on learning achievement and interest. In summary, making the right career decision is a primary goal for students. For this reason, many researchers have focused their attention on how to effectively mine the relations between an individual's personality and suitable career using subjective psychological tests, e.g., the Holland Code Career Test. With good test results, students will demonstrate better learning achievements for her/his potential career. In the remainder of this section, studies are reviewed on the basis of three categories defined in Figure 2, namely, "Personality and Career", "Personality and Learning" and "Technology and Learning".

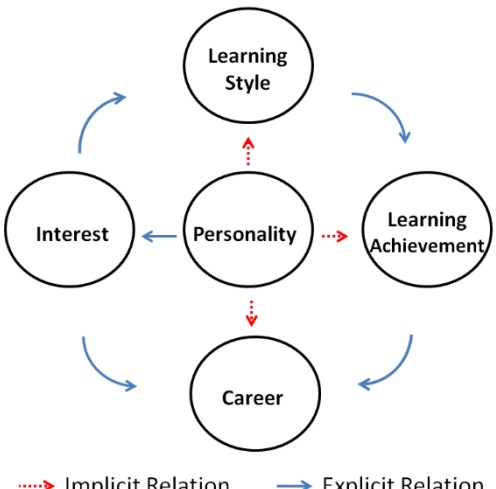

**Figure 2.** Relations between personality, interest, learning, achievement and career.

### 2.1. Personality and Career

As there is a strong association between personality and career, the topic of linking the two has been researched for a long time [24]. In this field, the Holland Code Career Test is popular and widely used to exploit the potential personality, interest and career [25]. Indeed, in most high-schools, students are given similar psychological tests, which are used to recommend an appropriate future career. For this purpose, Ramadhani et al. [26] provided Holland theory career guidance services for student career guidance and counseling. Their results show that Holland code testing works well. Ahmed et al. [27] reported experimental results that demonstrated the significant impact of Holland's RIASEC Scale test on occupational preferences. Budiyono et al. [28] constructed Web instrument products that were helpful in identifying potential career interests. Furthermore, Ayriza et al. [29] pointed out that, for children, the sociality in Holland theory is very relevant to the learning orientation to acquire knowledge and form preferences. Rocconi et al. [30] referred to

Holland theory to analyze the impact of person–environment fit on grades, perceived gains and satisfaction.

## 2.2. Personality and Learning

According to Figure 2, interest is related to personality. Furthermore, a strong interest yields effective learning and good achievements [31,32]. Interest can be regarded as a learning motivation. Without interest, learning might be boring. Cohen et al. [33] verified the relation between personality and satisfaction based on the Big Five model. DeNovellis et al. [34] integrated information about personality, interests and cognition to classify 36 subgroups for solving numerical, verbal and spatial problems. Dordi et al. [35] discovered that, for adolescents, the relations between parenting styles and academic motivations (according to the Parental Authority Questionnaire, Big Five Inventory and Academic Motivation Scale) are positive. Preuß et al. [36] attempted to infer the impacts of a learner's personality and motivation on mediators based on the Big Five model. Seinmayr et al. [37] employed grit to predict learning achievement while controlling for prior school grades, Big Five personality traits, school engagement, etc. This study also controlled for other factors, such as intelligence and conscientiousness, and established constructs from the literature on motivation and engagement to infer the importance of grit. Brandt et al. [38] validated the relations between the Big Five personality traits and cognitive ability based on academic achievements. The results showed that personality predicted more differences in academic achievements than other factors.

In addition to learning motivation, another factor of learning is learning style. Students with different personalities will have varying learning styles. Hence, their learning performances will differ. This is the major reason why a large number of past studies have been devoted to investigating the relations between personality and learning style. An et al. [10] presented methods to explain and predict learning differences based on learning style theory. Keshavarz et al. [39] showed that personality and learning style are very important considerations for increasing motivation for blended learning. Vasileva-Stojanovska et al. [40] and Laryea et al. [41] confirmed that personality and learning style significantly affected academic achievement. Puji et al. [42] revealed that the dominant learning style of history education students in Indonesia is the Extrovert, Intuition, Thinking, and Judging (ENTJ) type based on the Myers Briggs Type Indicator (MBTI) model, which defines four dimensions of personality. Therefore, teachers are recommended to perform student-centered learning.

## 2.3. Technology and Learning

Chien et al. [43] attempted to discover the impacts of Collaborative Interactions (CIs) and Human–Computer Agent (HCA) and Human–Human (HH) interactions on the Collaborative Problem-Solving (CPS) performance of students. Lytras et al. [44] proposed the idea of transformative education based on the concept of total quality management in human resources. Cyber-physical systems, sentiment management and ubiquitous learning delivery were also integrated for further technology-enhancing education. Shen et al. [45] identified five main types of Technology-Enhanced Learning (TEL). They also presented visual analytics on TEL research and a direct citation network for the future development of the TEL research domain. Wong et al. [46] attempted to find the relationship between interests and mathematics achievements in a technology-enhanced learning environment in Malaysia. According to their results, the association between interest and mathematics achievements is unclear for those with higher performances, whereas it is well defined for those with low performances. To increase learning achievement, Su et al. [1] inferred potential interest by determining associations between personality and achievement. In this study, through course recommendation and grade prediction, the students were informed of their potentialities in specific future course subjects. With this knowledge, the right decisions on courses of interest can be made. In 2020, due to the COVID-19 pandemic, e-learning attracted much more attention. Without face-to-face classes, students were required to

learn online through fast-developing e-learning technologies. Saeed Al-Maroof et al. [47] performed an investigation showing that a technology's self-efficacy, ease of use and usefulness to teachers and students in university directly affected the intention to continue its use. Müller et al. [48] conducted in-depth interviews with 14 educators from a large university in Singapore. In this study, educators stated that the flexibility of e-learning allowed students to learn independently and further prompted teachers to reflect on how to improve their practice through e-learning. However, to satisfy diverse needs, e-learning has to include social, emotional, and cognitive components.

## 3. Proposed Method

### 3.1. Basic Notion

As we recall from Figure 1, learning can basically be divided into three stages: "What to learn", "How to learn" and "How to apply". Most past studies have focused on "How to learn" and "How to apply". However, "what to learn" can impact the learning performance ("how to learn") and future career ("how to apply") because of learning interest. Although a related work [1] confirmed this concept and further proposed a solution to this issue, several problems remain unresolved. To reveal the overall differences, we compare this related work and the proposed method in the following.

- In the compared work, personality was represented by a set of profiles, a set of preferences and a set of self-recognized traits. On the contrary, personality in this paper is defined by Holland codes. Overall, our intent was to identify personality from a psychological point of view instead of profiles, preferences and self-recognition.
- In the compared work, the recommended courses were limited to one domain, so-called single-domain learning, while those in this paper include courses that cross multiple domains and related jobs.
- In the compared work, the prediction result was a score on a scale of 1–5. In contrast, in this paper, the prediction result is the score in float format, ranging from 0 to 100. This can provide the student with more detailed differences in expected learning performance.

For the experiments, the proposed approach was evaluated using a larger dataset than that in the compared work. To test the proposed idea, the system was implemented by collecting data from 12 departments, in contrast to only one department in the compared work.

### 3.2. Framework of the Proposed System

As mentioned in the previous section, a number of issues in recent studies remain unresolved. These issues motivated us to propose an innovative recommender system for cross-domain learning from a psychological point of view. Figure 3 shows the framework of the proposed system, which can be divided into two phases, namely, offline preprocessing and online recommendation. A quick overview of the proposed framework is provided in the following.

- Offline preprocessing

The goal of this phase is to generate a user-to-user similarity database by computing students' (also called "user" in this paper) personality similarities. To this end, this phase can further be divided into two sub-phases: data collection and data engineering. The personalities and performances of cross-domain students are collected in the data collection step, while data engineering is the processing of data and computation of user similarities.

- Online recommendation

Once an active student logs onto this system, it will perform a course score prediction and Holland code assessment. For the course score prediction, similar users are grouped to

calculate the course grades. For the Holland code assessment, related jobs are also derived by computing the scores of six personality types. As user similarities have been prepared in the offline preprocessing phase, the prediction is fast without online computations. Finally, a cross-domain course list and a set of related jobs will be shown to the student.

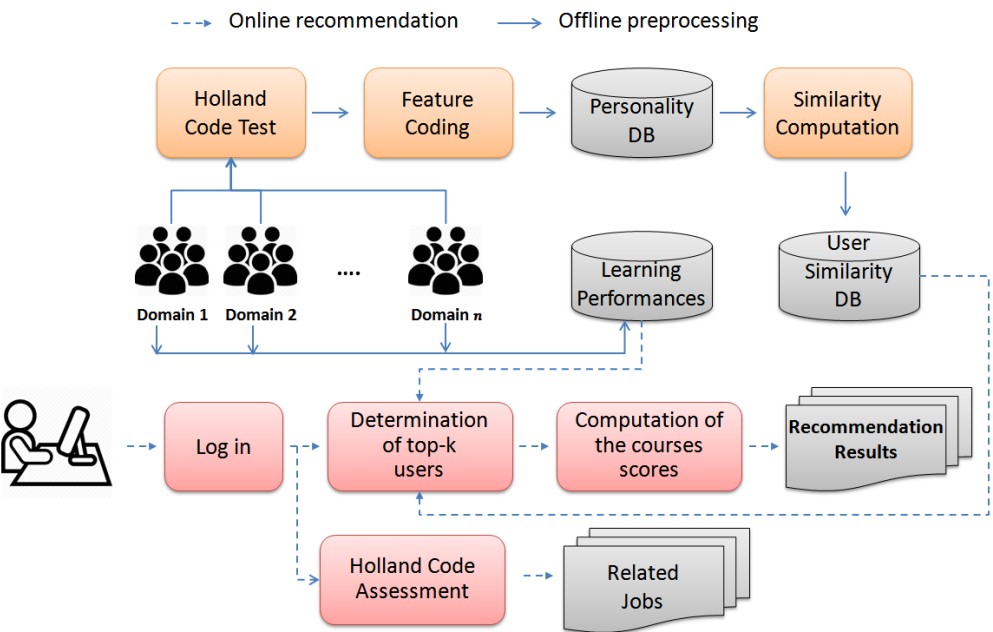

**Figure 3.** Framework of the proposed system.

### 3.3. Offline Preprocessing

The goal of this phase is to compute user-to-user similarities to speed up the online recommendation. The main processes, including data collection and data engineering, are explained in the following sub-sections.

#### 3.3.1. Data Collection

As this paper proposes a cross-domain recommender system, the first task is to gather the performances and personalities of cross-domain students. In this paper, a domain is a department in a university. Hence, it is necessary to gather students' performances and personalities from different departments. Performance is indicated by students' course scores after learning; the course scores are expressed as float type data, and the range is from 1 to 100. Therefore, a course-to-user score matrix is generated, which can be defined as

$$S_{C \rightarrow U}[s_{a,b}],$$

where $C$ and $U$ indicate sets of courses and students, respectively, and $s_{a,b}$ indicates the score of the $a$th course for the $b$th student.

In this paper, the student's personality is represented by Holland codes. For each student in different departments, she/he needs to complete the Holland code test. Holland codes define six personality types, namely, Realistic (R), Investigative (I), Artistic (A), Social (S), Enterprising (E) and Conventional (C). Each type consists of a set of options in the test, which is shown in Table 1.

#### 3.3.2. Data Engineering

After the student completes the Holland code test, the system transforms a term description into a binary code. There are 90 term descriptions is total in the test. Each description is converted into 1 if checked and 0 if unchecked. As a result, the personality is represented as a binary feature vector, which is defined in Definition 1.

**Table 1.** Holland code test.

| S1 | I1 | A1 | C2 |
|---|---|---|---|
| ☐Superiority complex | ☐I like complex things | ☐Difficult to understand | ☐Obstinate |
| ☐Cooperation | ☐Analytical | ☐Be out of order | ☐Overcautious |
| ☐Patient | ☐Cautious | ☐Emotional | ☐Regular |
| ☐Amiability | ☐Critical | ☐Expressive | ☐Obedient |
| ☐Generous | ☐Curious | ☐Idealize | ☐Methodic |
| **R2** | **A2** | **S2** | **I2** |
| ☐Materialism | ☐Intuitive | ☐Helpful | ☐Independence |
| ☐Spontaneous | ☐Independent | ☐Idealize | ☐Intellectual |
| ☐Ordinary | ☐Creative | ☐Merciful | ☐Soul-searching |
| ☐Perseverance | ☐Sensitive | ☐Persuasive | ☐Pessimistic |
| ☐Practical | ☐Open mindedness | ☐Put oneself in someone's shoes | ☐Precise |
| **R3** | **C3** | **I3** | **A3** |
| ☐Keep a low profile | ☐Perseverance | ☐Rational | ☐Imaginative |
| ☐Stubborn | ☐Practical | ☐Implied | ☐Impractical |
| ☐Simple | ☐Overmodest | ☐Conservative | ☐Impulsive |
| ☐The lack of insight | ☐Simple | ☐Overmodest | ☐Independence |
| ☐Bystander | ☐Lack imagination | ☐Unpopular | ☐Soul-searching |
| **S3** | **E2** | **E3** | **R1** |
| ☐Sesponsible | ☐Great vitality | ☐Romantic | ☐Imaginative |
| ☐Sociable | ☐Self-expression | ☐Optimistic | ☐Impractical |
| ☐Tact | ☐Thrill loving | ☐Confident | ☐Impulsive |
| ☐Understanding | ☐Impulsive | ☐Sociable | ☐Independence |
| ☐Warm | ☐Extrovert | ☐Talkative | ☐Soul-searching |
| **E1** | **C1** | | |
| ☐Indulge in profits and lust | ☐Cautiously | | |
| ☐Adventurous disposition | ☐Obedience | | |
| ☐Agreeable | ☐Righteous | | |
| ☐Ambitious | ☐Conservative | | |
| ☐Dominance | ☐Efficient | | |

**Definition 1.** *Given a set of personality description options $P = \{p_1, p_2, \ldots, p_{90}\}$, the personality feature vector for user $u_i$ is defined as*

$$F_i = \{f_{i,1}, f_{i,2}, \ldots, f_{i,90}\} \tag{1}$$

Then, the personality feature vectors of all students in different domains are collected. Based on the personality feature vectors, user-to-user similarity can be computed, which is defined in Definition 2.

**Definition 2.** *Given two personality feature vectors for users $u_i$ and $u_j$, the similarity between $u_i$ and $u_j$ is defined as*

$$sim(u_i, u_j) = \frac{\sum_{1 \leq x \leq 90} f_{i,x} * f_{j,x}}{\sqrt{\sum_{1 \leq x \leq i} f_{i,x}^2 * \sum_{1 \leq x \leq j} f_{j,x}^2 *}}. \tag{2}$$

Finally, all student similarities are calculated and then inserted into the user-to-user similarity matrix, which is defined as

$$M_{U \rightarrow U}[m_{i,j}],$$

where $U$ indicates the set of students, and $m_{i,j}$ indicates the similarity between the $i$th and $j$th students.

### 3.4. Online Recommendation

Based on the data processed in the offline phase, the online recommendation process is triggered when an active student logs onto the system. The process is described by the following steps.

Step 1: An active student logs onto the system. For a student using the system for the first time, she/he will be requested to complete the Holland code test before receiving recommendations.

Step 2: For an existing member, the system determines the top $k$ most similar students on the basis of user similarities in the similarity database. This operation is very fast because user similarities have been calculated in the offline phase.

Step 3: For each distinct course not being taken by the active student, the system is instructed to perform the following:

Step 3.1: Calculate the course score by considering the top $k$ most similar students using Definition 3;

Step 3.2: Continue until all non-major course scores are calculated, and then proceed to

Step 4: Threshold and rank the courses by the calculated scores.

Step 5: Return the ranking list to the active student.

**Definition 3.** *Given a course c to predict and k students who are similar to the active student $u_{act}$, the predicted score ŝ. of course c is defined as*

$$\hat{s}_{c,act} = \frac{\sum_{1 \le i \le k,\ s_{c,i} > 0} s_{c,i} * sim(u_i, u_{act})}{\sqrt{\sum_{1 \le i \le k,\ s_{c,i} > 0} sim(u_i, u_{act})}}, \tag{3}$$

*where $s_{c,i}$ indicates the course score of c for the ith student in the set of k similar students. In addition to the calculation of the course scores, another operation in the online phase is the Holland code assessment. For this operation, the scores of six personality types are computed by adding the options checked by the student. Finally, the jobs related to the top three personality types are presented to the active student.*

## 4. Experimental Settings

In the previous sections, the literature review, motivation and method are detailed. In the following sections, the experimental settings and evaluation analyses are shown in two aspects, namely, effectiveness evaluation (also called objective evaluation in this paper) and usage evaluation (also called subjective evaluation in this paper).

### 4.1. Experimental Environment and Data

To evaluate the proposed method, we constructed a system that recommends courses and jobs. The system was programmed using the Apache server, PHP and Java, and it was run on a PC with 64-bit Windows 7. After implementing the system, it was initialized using the experimental data. The experimental data were gathered from Cheng Shiu University, Taiwan, and the details of the data are shown in Table 2. From these data, 12 departments and 583 courses were used for experiments. Overall, 710 students were invited to complete the Holland code test, and 12,101 course grades for these 710 students were collected. From the 710 students, 293 students were selected for subjective evaluations, and 5 students per department were selected for objective evaluations; that is, 60 students were selected for objective evaluations in total.

**Table 2.** Details of experimental data.

| Data Parameter | Value |
|---|---|
| # Departments | 12 |
| # Courses | 583 |
| # Holland Code Tests | 710 |
| # Student Testers for Objective Evaluations | 60 |
| # Student Testers for Subjective Evaluations | 293 |
| # Course Scores | 12101 |

*4.2. Experimental Measures for Effectiveness Evaluation*

The effectiveness was evaluated using the metrics *Coverage* and *MAE* (Mean Absolute Error). The *Coverage* is defined as

$$Coverage = \frac{|Accurate|}{|Recommended|}, \tag{4}$$

where *Recommended* indicates the set of predicted courses, and *Accurate* indicates the correct prediction of a course set. Table 3 is an illustrative example of the evaluation measure *Coverage*. In this example, assume that there are five candidate courses and that the score threshold is 70. Then, the recommended set is {B, C, D} and the ground-truth set is {A, C, D} because the predicted and original scores, respectively, exceed 70. Therefore, the accurate set is {C, D}, and the *Coverage* is 2/3 = 67% accordingly.

**Table 3.** Example of the evaluation measure *Coverage*.

| Course | Predicted Scores | Original Scores | Recommended | Ground Truth | Accurate |
|---|---|---|---|---|---|
| A | 65 | 71 | | √ | |
| B | 72 | 68 | √ | | |
| C | 83 | 91 | √ | √ | √ |
| D | 91 | 88 | √ | √ | √ |
| E | 55 | 63 | | | |

In addition to coverage, the other measure in the objective evaluation is *MAE* (Mean Absolute Error), which can be defined as

$$MAE = \frac{\sum_{1 \leq i \leq |U|, \, 1 \leq c \leq |predicted_i|} \left| \hat{s}_{c,i} - s_{c,i} \right|}{|predicted_i|}, \tag{5}$$

where *U* indicates the set of student testers, $predicted_i$ indicates the set of unknown courses for the *i*th student, $\hat{s}_{c,i}$ indicates the predicted score, and $s_{c,i}$ indicates the original score for the *c*th unknown course in the $predicted_i$ set. For example, in Table 3, the *MAE* for {A, B, C, D} is (|65–71| + |72–68| + |83–91| + |91–88| + |55–63|)/5 = 5.8.

*4.3. Experimental Questionnaire for Usage Evaluation*

This subjective evaluation was performed after the testers used the proposed system. In this evaluation, there were four insights that we wanted to capture from the student testers, namely, (1) personal career, (2) the impact of course selection, (3) the current course selection system and (4) the usage satisfaction after using the proposed system, which were assessed from the questionnaire in Table 4. In this questionnaire, the student testers were requested to rate each question. The rating is an integer score ranging from 1 to 5, where 1, 2, 3, 4 and 5 denote "disagree", "somewhat disagree", "somewhat agree", "agree" and "agree very much", respectively. The 1st, 2nd and 3rd insights can be viewed as characterizing the motivation to use the proposed system, while the 4th one shows the usage satisfaction.

**Table 4.** Questionnaire for subjective evaluations.

| **Insight 1: Questions for the personal career** |
|---|
| 1.   I want to know my interests. |
| 2.   I clearly know what job is appropriate for me in the future. |
| 3.   I hope my job matches my interests. |
| **Insight 2: Questions for the impact of course selection** |
| 1.   It is necessary to provide cross-domain school courses that are helpful to a future job. |
| 2.   It is very important to select courses that match interests. |
| 3.   I want to know the course details related to my career interest. |
| 4.   It is difficult to make a choice when facing a number of cross-domain courses. |
| **Insight 3: Questions for the current course selection system** |
| 1.   The current course selection system cannot detect my interest. |
| 2.   The current course selection system cannot provide the service of cross-domain course selection. |
| 3.   It is important to provide the service of cross-domain course selection for a course selection system. |
| 4.   It is important to predict the unknown cross-domain course scores for a course selection system. |
| 5.   It is important to recommend a set of cross-domain courses related to my interest for a course selection system. |
| **Insight 4: Questions for usage satisfaction after using the proposed system** |
| 1.   The proposed system can provide cross-domain courses matching my interests. |
| 2.   The proposed system can precisely predict cross-domain course scores. |
| 3.   The proposed system can recommend cross-domain courses that are helpful to my career. |
| 4.   The proposed system is helpful to self-learning. |
| 5.   Overall, the proposed system can really help me select appropriate courses, including single- or cross-domain courses. |

## 5. Experimental Results

### 5.1. Results of Effectiveness Evaluations

This evaluation was mainly designed to test the proposed system using Coverage and *MAE*, which reveals how close the predicted results are to the ground-truth. Figure 4 shows the Coverage for different *k* similar students under different thresholds. From Figure 4, we can make two observations. First, the lower the threshold, the higher the coverage. Second, the higher the k-value, the lower the coverage. The unique findings and discussion are explained below.

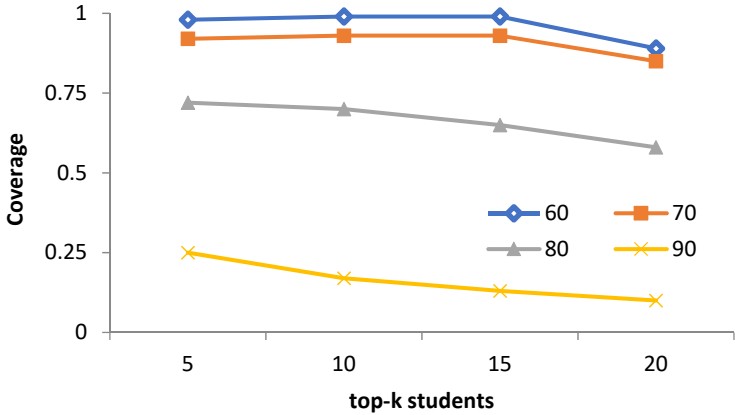

**Figure 4.** Coverage of the proposed system for the top *k* most similar students under thresholds of 60–90.

- The experimental results show that personality is diverse in students with higher course grades, especially for scores above 90. This can be explained by two concepts. First, the data show that very few students can achieve grades higher than 90. Hence, it is not easy to predict grades of 90 when there are very few known students with this grade. Second, students with similar personalities perform inconsistently for high course grades. However, the predicted scores are still close to the actual scores according to *MAE*.

- The coverage decreases as the value of *k* increases. This is because students that are less similar to the user are potential noises that skew the prediction.

Overall, the coverage shows the performance of course recommendations. To assess the prediction quality of the course score, we performed an additional experiment, namely, effectiveness evaluation using the mean absolute error. This metric is used because the coverage does not reveal the detailed differences between the predicted and original scores. To fill this gap, *MAE* was computed to provide the prediction quality in detail. Figure 5 shows the *MAE*s of the proposed system for the top *k* most similar students without thresholds. From these results, several observations can be made.

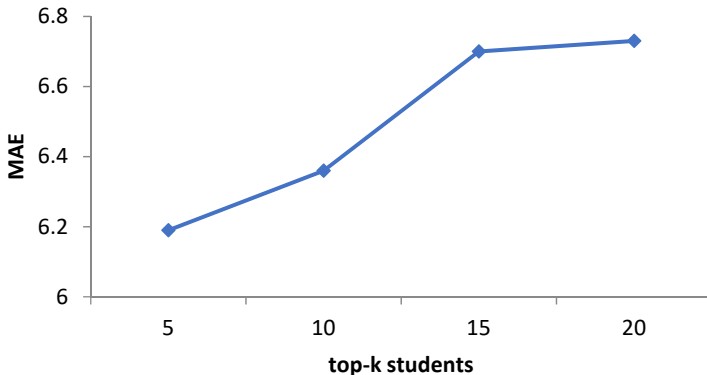

**Figure 5.** *MAE*s of the proposed system for the top *k* most similar students without thresholds.

- The results of these experiments can be viewed as additional support for the above finding. Using a larger number of similar students decreases the prediction quality because the scores of dissimilar students do not produce reliable score predictions.

- Although the coverage of high course grades is not high, the overall difference (*MAE*) is very small (around 6.19). This confirms that the proposed system performs well in predicting scores. This finding can be further verified in the following experiment, namely, usage evaluation.

### 5.2. Results of Usage Evaluations

The goal of this evaluation, which is detailed in Table 4, is to provide insights ranging from the motivation to usage satisfaction. These results were obtained using subjective questionnaires. Hence, the questionnaire provides four insights, namely, personal career, the impact of course selection, the current course selection system and the usage satisfaction after using the proposed system. For the first insight, Figure 6 shows the students' votes on their future career recommendation. These results can be summarized in the following points.

- Most of student testers wanted to know her/his interests. This is because they hoped that their interests would be suitable for their future job.

- Around 16.4% of student testers did not know which job was appropriate for their interests. Although university students are grouped by their selected interests in senior high schools in Taiwan, 16.4% of students were unsure.

- Around 97% of student testers agreed that person's interests should match her/his job. This is why they wanted to know their interests in the first question.

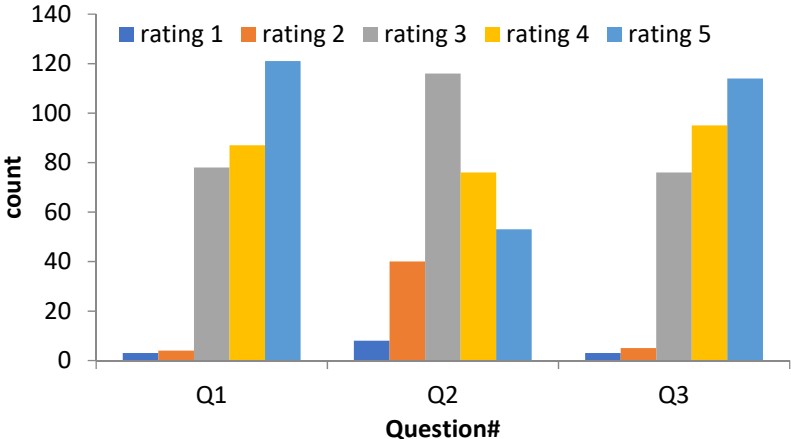

**Figure 6.** Experimental results for the first insight on usage evaluation.

For the second insight, Figure 7 shows the following:

- Around 91% of student testers needed to know which cross-domain courses would be helpful to them in their future jobs. This is because they had no idea how to select courses crossing domains that were pertinent to their interests.
- Most student testers agreed that matching courses with their interests is very important. Furthermore, they wanted to know the course details related to their future jobs. This evidence supports the study results presented in Section 2.
- However, when facing a number of cross-domain courses, around 87% of students encountered difficulties in choosing the appropriate ones. A potential explanation is that they were in departments that required one knowledge domain.

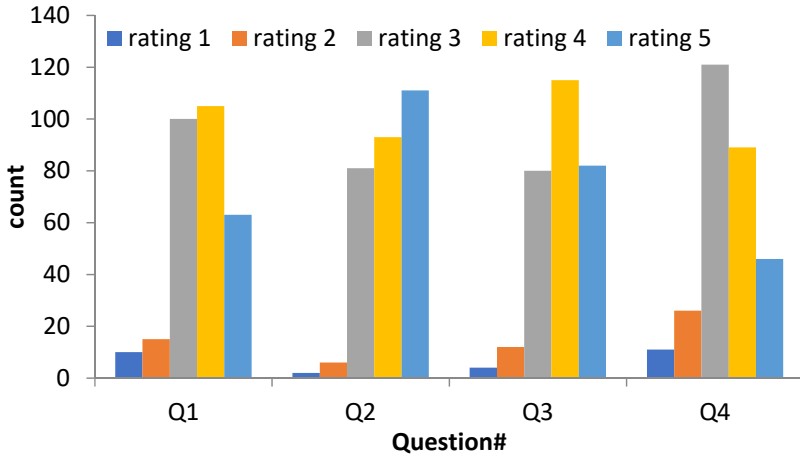

**Figure 7.** Experimental results for the second insight on usage evaluation.

To provide the third insight, the questions were designed to assess the current course selection system. These evaluation results are shown in Figure 8, and the findings are highlighted below:

- The results of the 1st and 2nd questions show that the current system cannot identify interests and cannot provide cross-domain recommendations. This reveals the shortcomings of the current system.
- As a result, around 96% of student testers needed a service to recommend relevant cross-domain courses that were suitable for their future jobs.
- Moreover, most of the student testers preferred being informed of their predicted scores for the recommended courses.

- The above findings show that most of the student testers were not satisfied with the current system because of the lack of course recommendations.

Overall, Figures 6–8 support the practicality of the proposed idea because most students needed an intelligent system to facilitate cross-domain course selection. Finally, Figure 9 depicts the results for our proposed system, which can be summarized as follows.

- Over 90% of students provided positive ratings for the 1st and 3rd questions, which suggests that the proposed system can identify personal interests from the Holland code assessment and further recommend courses that will be helpful to the student's future career.
- Over 90% of students agreed that the system was able to predict course scores and that the predicted scores were helpful for ensuring more effective learning.
- Overall, around 95.54% of students were satisfied with the proposed system in terms of its recommendation and prediction.

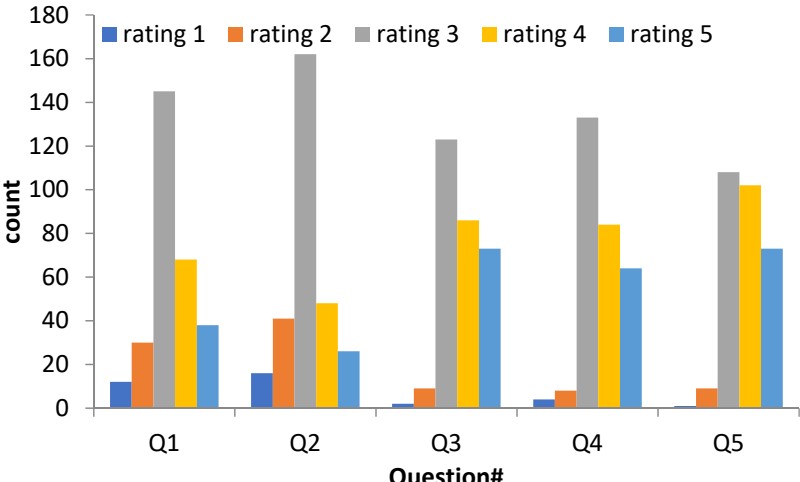

**Figure 8.** Experimental results for the third insight on usage evaluation.

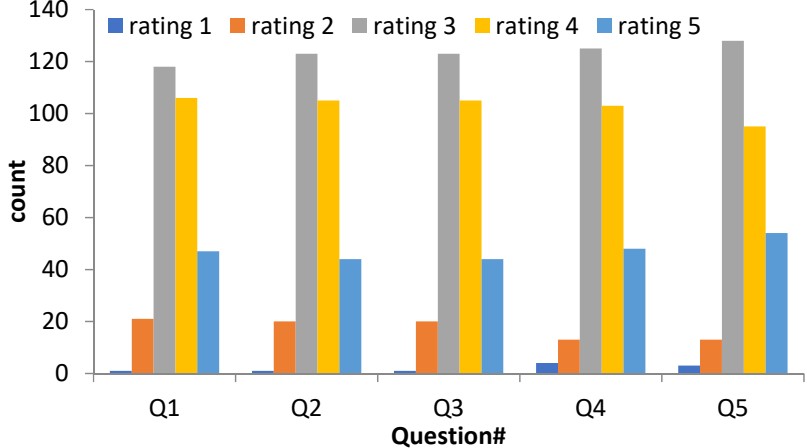

**Figure 9.** Experimental results of the fourth insight for usage evaluation.

## 6. Conclusions

Investigations into the association between personality and career have been performed for a long time. Although many previous methods, such as the Holland code test, have been shown to be effective in linking personality to career, very few studies have linked personality, learning and career. Modern job requirements have moved from

involving a single domain to multiple domains. Therefore, the need for cross-domain learning has also increased significantly in current universities. Unfortunately, no modern system has been proposed to meet this need. To fill this gap, in this paper, an intelligent course selection system is proposed to identify job interests, recommend courses and predict course scores. The Holland code assessment is employed to identify potential job interests, thereby providing students with knowledge of their appropriate jobs in the future. For the prediction of course scores, user-filtering is performed to group students with similar personalities. On the basis of this information, the unknown scores are calculated from the scores of similar students. Through the predicted scores, students can adjust their learning approach for courses with low predicted grades. For the recommendation of courses, based on the predicted scores, single- or cross-domain courses are shown to the students by using thresholds.

To evaluate the proposed idea, a system was developed by integrating the techniques mentioned in this paper. With the implemented system, effectiveness and usage evaluations were conducted from the viewpoints of course recommendations and user satisfaction. An experimental limitation is that the evaluation was performed in students from the same Taiwanese university, and the student testers were from 12 departments. The results of the objective evaluations show that the average prediction errors are very small (within 6.8), and most courses can be filtered correctly. This result shows the Holland code assessment is useful for linking interests and courses. Additionally, the results of the subjective evaluations show that most students do not know which cross-domain courses to select and would like to have a smart system for course selection. This motivated us to propose a personality-driven system. For objective evaluations, the results reveal that the proposed system can satisfy students' needs when choosing cross-domain courses. In the future, a number of issues will be explored further. First, more personality tests will be compared with the Holland code test to analyze their performances in predicting interests and scores. This is because other personality tests have been shown to be effective in defining personality. Second, the proposed recommender system will be tested in more universities. A difference analysis will be conducted to provide educators with useful information so as to increase learning interest and achievement. Third, more effective recommendation algorithms will be tested to improve the predictions. As mentioned in Section 2, numerous recommender methods have been proposed in different fields. In the future, better approaches will be tested to increase the quality of the recommendation. Fourth, in addition to personality, other useful information sources, such as social media, will be combined to improve the determination of students' interests. This is because a user's interest might be hidden in contextual information.

**Author Contributions:** Conceptualization, J.-H.S. and Y.-W.L.; methodology, J.-H.S. and Y.-W.L.; software, J.-Z.X.; validation, Y.-W.Z.; formal analysis, J.-H.S. and Y.-W.L.; investigation, J.-H.S. and Y.-W.L.; resources, J.-Z.X.; data curation, Y.-W.Z.; writing—original draft preparation, J.-H.S.; writing—review and editing, J.-H.S.; visualization, J.-Z.X.; supervision, J.-H.S. and Y.-W.L. All authors have read and agreed to the published version of the manuscript.

**Funding:** This research received funding from the Ministry of Science and Technology, Taiwan, under grant no. MOST 109-2511-H-230-001-MY2.

**Institutional Review Board Statement:** Not applicable.

**Informed Consent Statement:** Not applicable.

**Data Availability Statement:** Not applicable.

**Acknowledgments:** This article was supported by the Ministry of Science and Technology, Taiwan, under grant no. MOST 109-2511-H-230-001-MY2.

**Conflicts of Interest:** The authors declare no conflict of interest.

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
