# Peer review of "A Personality-Driven Recommender System for Cross-Domain Learning Based on Holland Code Assessments"

_sustainability, doi:10.3390/su13073936_

Round 1

Reviewer 1 Report

Dear Authors,

Please find below my recommendations an remarks both positive and negative regarding your manuscript proposal, which I hope will help you improve it.

The proposed paper is very interesting and the topic of recommender systems is very up-to-date, especially in the COVID times, where AI systems help to step in for human, also in the field of recommendations. The presented study suggests a novel “personality-driven” recommender system, in a way that it takes into account personality traits of users, to recommend cross-domain courses for students automatically. Computing personality similarities is performed based on that students similar to the active student are filtered by Holland-Code differentiations and the right course degrees are predicted by considering similar students.

From the technical side of manuscript preparation, the authors should note that in mdpi journals the citations should generally follow the ACS Style Guide. All references mentioned in the text should be numbered consecutively throughout the paper as they appear in the paper (starting from 1, which means that the Reference list needs to be rearranged). I

When citing more than one reference at one place, please list the numbers in ascending order and separate them by commas without spaces, or if they are in a consecutive series, use a dash to indicate a range of three or more, e.g. [5,6] or [7–9], instead of [5][6] and [7][8][9], respectively.

In the Abstract, I would recommend that you not only very clearly describe the goal of your research, but also mention the main implications and limitations. Also, it is advisable to make sure that the Abstract is understandable by any reader.

While the Previous Study section is well organized and thoughtful (maybe missing in s few more references to the most up-to-date papers), the Introduction section is somewhat failing in providing the full context. The first paragraph, lines 29-43, concentrates on AI use in education and learning domains. And then the second paragraph, lines 48-60, focuses on the aim of the paper and its uniqueness with regard to other papers. In this second paragraph I would recommend to include references supporting your important statements such as:

- “job is highly related to the learning”, line 52 – please refer to related studies, e.g. EJTD paper titled Factors influencing graduate job choice – a systematic literature review; GBMR paper titled Job selection preferences: what do young adults want; and APJSSR paper titled Factors that affects the job selection preference of undergraduate accounting students in university, etc.,

- “difference between this paper and the previous works”, line 55 – please refer to key previous works you had on your mind here,

- “the Holland codes”, line 58 – please refer to the right source so that readers unfamiliar with them could easily learn about them.

Furthermore, between paragraphs one and two, after line 46, I would recommend to provide additional content to give a more comprehensive context of recommender systems. A definition of a recommender system should first be provided (e.g. Springer Encyclopedia of Machine Learning: Recommender systems). After that, I would be glad to see a paragraph with a short introduction to most popular recommender techniques: collaborative filtering, content-based, demographic and knowledge-based – it was appropriately done in a recent Applied Sciences paper titled Horizontal vs. vertical recommendation zones evaluation using behavior tracking –concise descriptions from paragraph 4 can be utilized). Then, the increasing role of recommender systems due to their multi-domain applicability should be mentioned and the plentiful applications in providing personalized services should be emphasized with a number of up-to-date studies referenced (incl. Decision support systems paper titled Recommender system application developments: a survey; Procedia CS paper titled Human-website interaction monitoring in recommender systems; Concurrency and Computation paper titled Modeling online user product interest for recommender systems and ergonomics studies, LNDECT paper titled Evaluation of varying visual intensity and position of a recommendation in a recommending interface towards reducing habituation and improving sales). Finally, it would also be advisable to emphasize that recommender systems become increasingly quasi-intelligent thanks to the implementation of various artificial intelligence methods (adding current references, incl. Complex and Intelligent Systems paper titled Artificial intelligence in recommender systems; Electronics paper titled Deep learning-enhanced framework for performance evaluation of a recommending interface with varied recommendation position and intensity based on eye-tracking equipment data processing; Sensors paper titled Gaze and event tracking for evaluation of recommendation-driven purchase; and Smart Cities paper titled A predictive vehicle ride sharing recommendation system for smart cities commuting).

Like I mentioned before, the Previous Study section is quite good. It provides a critical appraisal of previous studies directly related to the paper. I would only suggest changing the title of the section to Previous Studies or Related Works, and adding more up-to-date sources dated 2020-21, in particular in the fast developing (also due to COVID) Technology and Learning field (Subsection 2.3), incl. Education Science paper titled The continuous intention to use e-learning from two different perspectives; and paper titled COVID-19 emergency elearning and beyond: experiences and perspectives of university educators, published in the same journal),

In Subsection 3.1, lines 148-168 I would refrain from quoting reference 27 as many as 8 times. I would recommend rewording the subsection informing that it is widely referring to the very good Applied Sciences paper titled An intelligent course decision assistant by mining and filtering learners’ personality patterns.

In Subsection 4.2.2, lines 333-334, I would recommend to provide the meaning of each individual rating 1-5, instead of just mentioning that 1-2 are negative and 3-5 are positive ratings. This is important since the ratings are used further in a number of figures.

Then, I would be happy to see a more in-depth discussion of the results shown in each of the Figures 6-9, perhaps split so that the comments are placed near the corresponding figure.

While I would leave all information on the experimental setting, questionnaire for subjective evaluations etc. in Section 4, I would recommend moving all results from this section to a new Section 5. Results.

In Conclusions, I advise to include the limitations of the experiment. Please provide a short summary of the findings, including brief evaluation of the system. I believe that readers would be happy to see there more future directions of your interesting research.

I also came across a number of grammar mistakes, so a language check should be done later if your paper is accepted after corrections. Anyway, a good proof-reading of the whole paper is recommended.

Please consider the above recommendations as being constructive remarks in order to improve the general quality of your interesting manuscript proposal.  

Good luck in your research work!

Author Response

The authors are grateful for the reviewers’ helpful comments that are valuable in improving this paper. We have revised the manuscript as follows. Or, please find the attached file for a clear review.

Revision made in accordance with comments by Reviewer No.1

  1. From the technical side of manuscript preparation, the authors should note that in mdpi journals the citations should generally follow the ACS Style Guide. All references mentioned in the texts should be numbered consecutively throughout the paper as they appear in the paper (starting from 1, which means that the Reference list needs to be rearranged). When citing more than one reference at one place, please list the numbers in ascending order and separate them by commas without spaces, or if they are in a consecutive series, use a dash to indicate a range of three or more, e.g. [5,6] or [7–9], instead of [5][6] and [7][8][9], respectively..

Answer: Thanks for this comment. We have re-sorted the reference list and modified the citations using the right format.

  1. In the Abstract, I would recommend that you not only very clearly describe the goal of your research, but also mention the main implications and limitations. Also, it is advisable to make sure thatthe Abstract is understandable by any reader.

Answer: Thanks for this comment. We have rewritten the abstract as follows based on the suggestion.

Abstract: Over the past few decades, AI has been widely used in the field of education. However, very little attention has been paid to the use of AI for enhancing the quality of cross-domain learning. College/university students are often interested in different domains of knowledge but may be unaware of how to choose relevant cross-domain courses. Therefore, this paper presents a personality-driven recommender system that suggests cross-domain courses and related jobs by computing personality similarities and probable course grades. In this study, 710 students from 12 departments in a Taiwanese university conducted Holland code assessments. Based on the assessments, a comprehensive empirical study, including objective and subjective evaluations, was performed. The results reveal that (1) the recommender system shows very promising performances in predicting course grades (objective evaluations), (2) most of the student testers had encountered difficulties in selecting cross-domain courses and needed the further support of a recommender system, and (3) most of the student testers positively rated the proposed system (subjective evaluations). In summary, Holland code assessments are useful for connecting personalities, interests and learning styles, and the proposed system provides helpful information that supports good decision-making when choosing cross-domain courses.

  1. While the Previous Study section is well organized and thoughtful (maybe missing in s few more references to the most up-to-date papers), the Introduction section is somewhat failing in providing the full context. The first paragraph, lines 29-43, concentrates on AI use in education and learning domains. And then the second paragraph, lines 48-60, focuses on the aim of the paper and its uniqueness with regard to other papers. In this second paragraph I would recommend to include references supporting your important statements such as:
  • “job is highly related to the learning”, line 52 – please refer to related studies, e.g. EJTD paper titled Factors influencing graduate job choice – a systematic literature review; GBMR paper titled Job selection preferences: what do young adults want; and APJSSR paper titled Factors that affects the job selection preference of undergraduate accounting students in university, etc.,

Answer: Thanks for this comment. We have added the related works [17-19] into the reference list and modified the citations in Page 2.

  • “difference between this paper and the previous works”, line 55 –please refer to key previous works you had on your mind here,

Answer: Thanks for this comment. We have added the citations [1, 20] in Page 2.

  • “the Holland codes”, line 58 – please refer to the right source so that readers unfamiliar with them could easily learn about them.

Answer: Thanks for this comment. We have added related work [21] into the reference list and modified the citations in Page 2.

  1. Furthermore, between paragraphs one and two, after line 46, I would recommend to provide additional content to give a more comprehensive context of recommender systems.

Answer: Thanks for this comment. Based on the comment, a new paragraph was added between the first and second paragraphs in Page 2, as shown bellow. 

When students need guidance to determine what to learn from multiple domains, a recommender system can serve as an effective solution by providing them with useful information. In principle, a recommender system uses a set of learning algorithms to discover user preferences to yield useful recommendations [5]. Generally, user preferences are represented by two types, namely, explicit and implicit preferences. Explicit preferences are identified from a user’s ratings of items with scores of 1-5, where 1 and 2 indicate a negative answer and 3, 4 and 5 indicate an affirmative response. A user’s implicit preferences are hidden in their behaviors, such as social media tags, navigation logs, browsing history and so on. Therefore, in this field, determining how to effectively discover the user-to-item affinity based on an individual’s preferences has become a challenging issue in recent years. To explore this issue, a number of recent recommender techniques have been proposed, including collaborative filtering, emotional-based, content-based, demographic and knowledge-based recommender systems [6-8]. With these techniques, in the past few years, recommender systems have been adopting an increasingly prominent role because of their multi-domain applicability and the abundance of applications that provide personalized services [9-12]. Thus, recommender systems can be regarded as being due to the implementation of various artificial intelligence methods [13-16].

  • A definition of a recommender system should first be provided (e.g. Springer Encyclopedia of Machine Learning: Recommender systems).

Answer: Thanks for this comment. We have added the related work [5] into the reference list and defined the recommender system in Line 52 of Page 2. Please refer to the following description for a quick review.

In principle, a recommender system uses a set of learning algorithms to discover user preferences to yield useful recommendations [5].

  • After that, I would be glad to see a paragraph with a short introduction to most popular recommender techniques: collaborative filtering, content-based, demographic and knowledge-based – it was appropriately done in a recent Applied Sciences paper titled Horizontal vs. vertical recommendation zones evaluation using behavior tracking –concise descriptions from paragraph 4 can be utilized).

Answer: Thanks for this comment. We have added the description and related work [6] into Lines 57-61 of Page 2. Please refer to the following description for a quick review.

Therefore, in this field, determining how to effectively discover the user-to-item affinity based on an individual’s preferences has become a challenging issue in recent years. To explore this issue, a number of recent recommender techniques have been proposed, including collaborative filtering, emotional-based, content-based, demographic and knowledge-based recommender systems [6-8].

  • Then, the increasing role of recommender systems due to their multi-domain applicability should be mentioned and the plentiful applications in providing personalized services should be emphasized with a number of up-to-date studies referenced (incl. Decision support systems paper titled Recommender system application developments: a survey; Procedia CS paper titled Human-website interaction monitoring in recommender systems; Concurrency and Computation paper titled Modeling online user product interest for recommender systems and ergonomics studies, LNDECT paper titled Evaluation of varying visual intensity and position of a recommendation in a recommending interface towards reducing habituation and improving sales).

Answer: Thanks for this comment. We have added related works [9-12] into the reference list and modified the citations in Page 2.

  • Finally, it would also be advisable to emphasize that recommender systems become increasingly quasi-intelligent thanks to the implementation of various artificial intelligence methods (adding current references, incl. Complex and Intelligent Systems paper titled Artificial intelligence in recommender systems; Electronics paper titled Deep learning-enhanced framework for performance evaluation of are commending interface with varied recommendation position and intensity based on eye-tracking equipment data processing; Sensors paper titled Gaze and event tracking for evaluation of recommendation-driven purchase; and Smart Cities paper titled A predictive vehicle ride sharing recommendation system for smart cities commuting).

Answer: Thanks for this comment. We have added related works [13-16] into the reference list and modified the citations in Page 2.

  1. Like I mentioned before, the Previous Study section is quite good. It provides a critical appraisal of previous studies directly related to the paper. I would only suggest changing the title of the section to Previous Studies or Related Works, and adding more up-to-date sources dated 2020-21, in particular in the fast developing (also due to COVID) Technology and Learning field (Subsection 2.3), incl. Education Science paper titled The continuous intention to use e-learning from two different perspectives; and paper titled COVID-19 emergency elearning and beyond: experiences and perspectives of university educators, published in the same journal)

Answer: Thanks for this comment. The title of Section 2 has been updated as “Previous Studies”. According to this comment, we added descriptions into sub-Section 2.3 (Lines 170-178) as follows and also added the citations [47-48] in Page 5.

In 2020, due to the COVID-19 pandemic, e-learning attracted much more attention. Without face-to-face classes, students were required to learn online through fast-developing e-learning technologies. Saeed Al-Maroof et al. [47] performed an investigation showing that a technology’s self-efficacy, ease of use and usefulness to teachers and students in university directly affected the intention to continue its use. Müller et al. [48] conducted in-depth interviews with 14 educators from a large university in Singapore. In this study, educators stated that the flexibility of e-learning allowed students to learn independently and further prompted teachers to reflect on how to improve their practice through e-learning. However, to satisfy diverse needs, e-learning has to include social, emotional, and cognitive components.

  1. In Subsection 3.1, lines 148-168 I would refrain from quoting reference 27 as many as 8 times. I would recommend rewording the subsection informing that it is widely referring to the very good Applied Sciences paper titled An intelligent course decision assistant by mining and filtering learners’ personality patterns.

Answer: Thanks for this comment. Based on the comment, the paragraph in Page 5 has been modified as follows.

As we recall from Figure 1, learning can basically be divided into three stages: “What to learn”, “How to learn” and “How to apply”. Most past studies have focused on “How to learn” and “How to apply”. However, “what to learn” can impact the learning performance (“how to learn”) and future career (“how to apply”) because of learning interest. Although a related work [1] confirmed this concept and further proposed a solution to this issue, several problems remain unresolved. To reveal the overall differences, we compare this related work and the proposed method in the following.

  • In the compared work, personality was represented by a set of profiles, a set of preferences and a set of self-recognized traits. On the contrary, personality in this paper is defined by Holland codes. Overall, our intent was to identify personality from a psychological point of view instead of profiles, preferences and self-recognition.
  • In the compared work, the recommended courses were limited to one domain, so-called single-domain learning, while those in this paper include courses that cross multiple domains and related jobs.
  • In the compared work, the prediction result was a score on a scale of 1–5. In contrast, in this paper, the prediction result is the score in float format, ranging from 0 to 100. This can provide the student with more detailed differences in expected learning performance.
  • For the experiments, the proposed approach was evaluated using a larger dataset than that in the compared work. To test the proposed idea, the system was implemented by collecting data from 12 departments, in contrast to only one department in the compared work.

  1. In Subsection 4.2.2, lines 333-334, I would recommend to provide the meaning of each individual rating 1-5, instead of just mentioning that 1-2 are negative and 3-5 are positive ratings. This is important since the ratings are used further in a number of figures.

Answer: Thanks for this comment. Yes, the definition is not clear. For this problem, in Lines 327-329 of Page 10, we redefined them by the description of “The rating is an integer score ranging from 1 to 5, where 1, 2, 3, 4 and 5 denote “disagree”, “somewhat disagree”, “somewhat agree”, “agree” and “agree very much”, respectively.”

  1. Then, I would be happy to see a more in-depth discussion of the results shown in each of the Figures 6-9, perhaps split so that the comments are placed near the corresponding figure. While I would leave all information on the experimental setting, questionnaire for subjective evaluations etc. in Section 4, I would recommend moving all results from this section to a new Section 5.Results

Answer: Thanks for this comment. For this comment, first, we reset the Sections 4 and split the experimental results into a new Section 5. Second, a detailed analysis for Figures 6-9 was added in Section 5. Please find the modifications from Pages 11-13 and refer to follows for a quick review.

The goal of this evaluation, which is detailed in Table 4, is to provide insights ranging from the motivation to usage satisfaction. These results were obtained using subjective questionnaires. Hence, the questionnaire provides four insights, namely, personal career, the impact of course selection, the current course selection system and the usage satisfaction after using the proposed system. For the first insight, Figure 6 shows the students’ votes on their future career recommendation. These results can be summarized in the following points.

Figure 6. Experimental results for the first insight on usage evaluation.

  • Most of student testers wanted to know her/his interests. This is because they hoped that their interests would be suitable for their future job.
  • Around 16.4% of student testers did not know which job was appropriate for their interests. Although university students are grouped by their selected interests in senior high schools in Taiwan, 16.4% of students were unsure.
  • Around 97% of student testers agreed that person’s interests should match her/his job. This is why they wanted to know their interests in the first question.

For the second insight, Figure 7 shows the following:

  • Around 91% of student testers needed to know which cross-domain courses would be helpful to them in their future jobs. This is because they had no idea how to select courses crossing domains that were pertinent to their interests.
  • Most student testers agreed that matching courses with their interests is very important. Furthermore, they wanted to know the course details related to their future jobs. This evidence supports the study results presented in Section 2.
  • However, when facing a number of cross-domain courses, around 87% of students encountered difficulties in choosing the appropriate ones. A potential explanation is that they were in departments that required one knowledge domain.

Figure 7. Experimental results for the second insight on usage evaluation.

To provide the third insight, the questions were designed to assess the current course selection system. These evaluation results are shown in Figure 8, and the findings are highlighted below:

  • The results of the 1st and 2nd questions show that the current system cannot identify interests and cannot provide cross-domain recommendations. This reveals the shortcomings of the current system.
  • As a result, around 96% of student testers needed a service to recommend relevant cross-domain courses that were suitable for their future jobs.
  • Moreover, most of the student testers preferred being informed of their predicted scores for the recommended courses.
  • The above findings show that most of the student testers were not satisfied with the current system because of the lack of course recommendations.

Overall, Figures 6-8 support the practicality of the proposed idea because most students needed an intelligent system to facilitate cross-domain course selection. Finally, Figure 9 depicts the results for our proposed system, which can be summarized as follows. 

  • Over 90% of students provided positive ratings for the 1st and 3rd questions, which suggests that the proposed system can identify personal interests from the Holland code assessment and further recommend courses that will be helpful to the student’s future career.
  • Over 90% of students agreed that the system was able to predict course scores and that the predicted scores were helpful for ensuring more effective learning.
  • On the whole, around 95.54% of students were satisfied with the proposed system in terms of its recommendation and prediction.

Figure 8. Experimental results for the third insight on usage evaluation.

Figure 9. Experimental results of the fourth insight for usage evaluation.

  1. In Conclusions, I advise to include the limitations of the experiment. Please provide a short summary of the findings, including brief evaluation of the system. I believe that readers would be happy to see there more future directions of your interesting research.

Answer: Thanks for this comment. As suggested, we have modified the conclusion. First, we added the experimental limitations. Second, we summarized the experimental results for subjective and objective evaluations. Third, we clearly showed the future works. Please refer to the following for a quick review.

To evaluate the proposed idea, a system was developed by integrating the techniques mentioned in this paper. With the implemented system, effectiveness and usage evaluations were conducted from viewpoints of course recommendations and user satisfaction. The limitation is the evaluation was performed in students from the same Taiwanese university, and the student testers were from 12 departments. The results of the objective evaluations show that the average prediction errors are very small (within 6.8), and most courses can be filtered correctly. This result shows the Holland code assessment is useful for linking interests and courses. Also the results of the subjective evaluations show that most students do not know which cross-domain courses to select and would like to have a smart system for course selection. This motivated us to propose a personality-driven system. For objective evaluations, the results reveal that the proposed system can satisfy students’ needs when choosing cross-domain courses. In the future, a number of issues will be explored further. First, more personality tests will be compared with the Holland code test to analyze their performances in predicting interests and scores. This is because other personality tests have been shown to be effective in defining personality. Second, the proposed recommender system will be tested in more universities. A difference analysis will be conducted to provide educators with useful information so as to increase learning interest and achievement. Third, more effective recommendation algorithms will be tested to improve the predictions. As mentioned in Section 2, numerous recommender methods have been proposed in different fields. In the future, better approaches will be tested to increase the quality of the recommendation. Fourth, in addition to personality, other useful information sources, such as social media, will be combined to improve the determination of students’ interests. This is because a user’s interest might be hidden in contextual information.

  1. I also came across a number of grammar mistakes, so a language check should be done later if your paper is accepted after corrections. Anyway, a good proof-reading of the whole paper is recommended.

Answer: Thanks for this comment. For this comment, we used the MDPI editing service to enhance the writing quality. Please refer to the following Certificate.

Reviewer 2 Report

The paper considers interesting and up-to-date topic as it considers achieving high-quality of cross-domain learning by artificial intelligence. Artificial Intelligence used for education has been studied for lots of years and  has been widely used in the field of education over the past few decades. Yet, very little attention was paid on how to achieve high-quality of cross-domain learning by AI. Although the investigation of bridging the learning interest to personality has been made by many research works, it cannot still cater to the cross-domain learning demands. This is because there exist multiple domain interests for a student in real applications.  In college/university, a student might be interested in different domains of knowledge but have no idea for choosing the related cross-domain courses. For this need, in this paper, a personality-driven recommender system is proposed to recommend the cross-domain courses and the related jobs by computing the personality similarities and course degrees

The aim of the research refers to pracitical needs. As a result of the paper a good recommendation to guide the students towards the  interest area of crossing multiple domains in addition to a single domain were elaborated. I find the solution proposed by the authors positively and practical useful. Future research opportunities and areas were not indicated.

The experimental results reveal that, most of the testing students agree with the proposed system can provide helpful information in making good decisions on cross-domain courses.  The paper was carefully edited. It needs only some editorial corrections and minor spell check are required. The English language and style are proper.

Please, add the limitations of the research and introduce some future research areas. Minor spell check are required.

Author Response

The authors are grateful for the reviewers’ helpful comments that are valuable in improving this paper. We have revised the manuscript as follows. Or, please find the attached file for a clear review.

Revision made in accordance with comments by Reviewer No.2

  1. The aim of the research refers to practical needs. As a result of the paper a good recommendation to guide the students towards the interest area of crossing multiple domains in addition to a single domain were elaborated. I find the solution proposed by the authors positively and practical useful. Future research opportunities and areas were not indicated. The experimental results reveal that, most of the testing students agree with the proposed system can provide helpful information in making good decisions on cross-domain courses. The paper was carefully edited. It needs only some editorial corrections and minor spell check are required. The English language and style are proper. Please, add the limitations of the research and introduce some future research areas. Minor spell check is required.

Answer: Thanks for this comment. For this comment, we used the MDPI editing service to enhance the writing quality. Please refer to the following Certificate. Also the limitations were added into the abstract and final section. Moreover, the future work has been updated as suggested. Please refer to the following for a quick review.

In Abstract:

In this study, 710 students from 12 departments in a Taiwanese university conducted Holland code assessments.

In Pages 13-14:

To evaluate the proposed idea, a system was developed by integrating the techniques mentioned in this paper. With the implemented system, effectiveness and usage evaluations were conducted from viewpoints of course recommendations and user satisfaction. The experimental limitation is the evaluation was performed in students from the same Taiwanese university, and the student testers were from 12 departments. The results of the objective evaluations show that the average prediction errors are very small (within 6.8), and most courses can be filtered correctly. This result shows the Holland code assessment is useful for linking interests and courses. Also the results of the subjective evaluations show that most students do not know which cross-domain courses to select and would like to have a smart system for course selection. This motivated us to propose a personality-driven system. For objective evaluations, the results reveal that the proposed system can satisfy students’ needs when choosing cross-domain courses. In the future, a number of issues will be explored further. First, more personality tests will be compared with the Holland code test to analyze their performances in predicting interests and scores. This is because other personality tests have been shown to be effective in defining personality. Second, the proposed recommender system will be tested in more universities. A difference analysis will be conducted to provide educators with useful information so as to increase learning interest and achievement. Third, more effective recommendation algorithms will be tested to improve the predictions. As mentioned in Section 2, numerous recommender methods have been proposed in different fields. In the future, better approaches will be tested to increase the quality of the recommendation. Fourth, in addition to personality, other useful information sources, such as social media, will be combined to improve the determination of students’ interests. This is because a user’s interest might be hidden in contextual information.

Reviewer 3 Report

The authors designed a personality recommender systems to recommend the cross-domain courses and the related jobs by computing the personality similarities and course degrees.

The proposed study is interesting but there are some points that the authors should better discuss.

The Abstract looks more like an introduction. It does not provide a brief of the methodological details neither the results of the experiments conducted for the purposes of this study. 

The authors should be better described the novelties of their study with respect to existing ones. In particular, the author should discuss limitation and cons of the examined approaches. Furthermore, the authors should provide more details and discussion about the obtained results. In particular, the authors should provide more discussion about Figure 6-9. The Discussion section also needs to be improved by analyzing the outcome of evaluation section.

I suggest to further analyze more recent approaches about the examined topics. In particular, I suggest the following papers to investigate scientific community based approaches and emotional state in the introduction:

1) An emotional recommender system for music. IEEE Intelligent Systems.

2) Kira: a system for knowledge-based access to multimedia art collections. In 2017 IEEE 11th international conference on semantic computing (ICSC) (pp. 338-343). IEEE.

Finally, I suggest to perform a linguistic revision.

Author Response

The authors are grateful for the reviewers’ helpful comments that are valuable in improving this paper. We have revised the manuscript as follows. Or, please find the attached file for a clear review.

Revision made in accordance with comments by Reviewer No.3

  1. The Abstract looks more like an introduction. It does not provide a brief of the methodological details neither the results of the experiments conducted for the purposes of this study.

Answer: Thanks for this comment. We have rewritten the abstract as follows based on the suggestion.

Abstract: Over the past few decades, AI has been widely used in the field of education. However, very little attention has been paid to the use of AI for enhancing the quality of cross-domain learning. College/university students are often interested in different domains of knowledge but may be unaware of how to choose relevant cross-domain courses. Therefore, this paper presents a personality-driven recommender system that suggests cross-domain courses and related jobs by computing personality similarities and probable course grades. In this study, 710 students from 12 departments in a Taiwanese university conducted Holland code assessments. Based on the assessments, a comprehensive empirical study, including objective and subjective evaluations, was performed. The results reveal that (1) the recommender system shows very promising performances in predicting course grades (objective evaluations), (2) most of the student testers had encountered difficulties in selecting cross-domain courses and needed the further support of a recommender system, and (3) most of the student testers positively rated the proposed system (subjective evaluations). In summary, Holland code assessments are useful for connecting personalities, interests and learning styles, and the proposed system provides helpful information that supports good decision-making when choosing cross-domain courses.

  1. The authors should be better described the novelties of their study with respect to existing ones.

Answer: Thanks for this comment. The main uniqueness of this paper is described in Page 2 and Page 5. Please refer to the following for a quick review.

In Page 2:

In the method proposed in this paper, the aim is to provide good recommendations to guide students towards their interest areas in multiple domains in addition to a single domain. The primary innovation of this method over traditional learning systems is three-fold.

  • In terms of learning stages, the proposed method focuses on the first stage of “what to learn” instead of “how to learn” and “how to apply”. This is because a person’s job is highly related to her/his learning direction [17-19], and the learning direction is implied by the learning interest. Therefore, our intent is to leverage the learning interest to determine what the student needs to learn.
  • In terms of what to learn, the major difference between this paper and previous works [1, 20] is that the proposed system aims to provide useful recommendations when a student faces a number of cross-domain courses. With this information, cross-domain learning achievements can be significantly enhanced.
  • In terms of discovering cross-domain interests, in the proposed approach, Holland codes [21] are used as personality features and form the basis on which personality similarities are computed. According to these similarities, the student’s potential grades in cross-domain courses of interest are predicted.

In Page 5:

As we recall from Figure 1, learning can basically be divided into three stages: “What to learn”, “How to learn” and “How to apply”. Most past studies have focused on “How to learn” and “How to apply”. However, “what to learn” can impact the learning performance (“how to learn”) and future career (“how to apply”) because of learning interest. Although a related work [1] confirmed this concept and further proposed a solution to this issue, several problems remain unresolved. To reveal the overall differences, we compare this related work and the proposed method in the following.

  • In the compared work, personality was represented by a set of profiles, a set of preferences and a set of self-recognized traits. On the contrary, personality in this paper is defined by Holland Overall, our intent was to identify personality from a psychological point of view instead of profiles, preferences and self-recognition.
  • In the compared work, the recommended courses were limited to one domain, so-called single-domain learning, while those in this paper include courses that cross multiple domains and related jobs.
  • In the compared work, the prediction result was a score on a scale of 1–5. In contrast, in this paper, the prediction result is the score in float format, ranging from 0 to 100. This can provide the student with more detailed differences in expected learning performance.
  • For the experiments, the proposed approach was evaluated using a larger dataset than that in the compared work. To test the proposed idea, the system was implemented by collecting data from 12 departments, in contrast to only one department in the compared work.

  1. In particular, the author should discuss limitation and cons of the examined approaches.

Answer: Thanks for this comment. We have added limitations into Pages 1, 13 and 14. Moreover the remaining issues (cons) of the proposed approach are shown in Page 14. Please refer to the following for a quick review.

In Abstract:

In this study, 710 students from 12 departments in a Taiwanese university conducted Holland code assessments.

In Pages 13-14:

To evaluate the proposed idea, a system was developed by integrating the techniques mentioned in this paper. With the implemented system, effectiveness and usage evaluations were conducted from viewpoints of course recommendations and user satisfaction. The experimental limitation is the evaluation was performed in students from the same Taiwanese university, and the student testers were from 12 departments. The results of the objective evaluations show that the average prediction errors are very small (within 6.8), and most courses can be filtered correctly. This result shows the Holland code assessment is useful for linking interests and courses. Also the results of the subjective evaluations show that most students do not know which cross-domain courses to select and would like to have a smart system for course selection. This motivated us to propose a personality-driven system. For objective evaluations, the results reveal that the proposed system can satisfy students’ needs when choosing cross-domain courses. In the future, a number of issues will be explored further. First, more personality tests will be compared with the Holland code test to analyze their performances in predicting interests and scores. This is because other personality tests have been shown to be effective in defining personality. Second, the proposed recommender system will be tested in more universities. A difference analysis will be conducted to provide educators with useful information so as to increase learning interest and achievement. Third, more effective recommendation algorithms will be tested to improve the predictions. As mentioned in Section 2, numerous recommender methods have been proposed in different fields. In the future, better approaches will be tested to increase the quality of the recommendation. Fourth, in addition to personality, other useful information sources, such as social media, will be combined to improve the determination of students’ interests. This is because a user’s interest might be hidden in contextual information.

  1. Furthermore, the authors should provide more details and discussion about the obtained results. In particular, the authors should provide more discussion about Figure 6-9. The Discussion section also needs to be improved by analyzing the outcome of evaluation section.

Answer: Thanks for this comment. For this comment, a detailed analysis for Figures 6-9 was added into a new Section 5. Please find the modifications from Page 11 to Page 13 and refer to the following for a quick review.

The goal of this evaluation, which is detailed in Table 4, is to provide insights ranging from the motivation to usage satisfaction. These results were obtained using subjective questionnaires. Hence, the questionnaire provides four insights, namely, personal career, the impact of course selection, the current course selection system and the usage satisfaction after using the proposed system. For the first insight, Figure 6 shows the students’ votes on their future career recommendation. These results can be summarized in the following points.

Figure 6. Experimental results for the first insight on usage evaluation.

  • Most of student testers wanted to know her/his interests. This is because they hoped that their interests would be suitable for their future job.
  • Around 16.4% of student testers did not know which job was appropriate for their interests. Although university students are grouped by their selected interests in senior high schools in Taiwan, 16.4% of students were unsure.
  • Around 97% of student testers agreed that person’s interests should match her/his job. This is why they wanted to know their interests in the first question.

For the second insight, Figure 7 shows the following:

  • Around 91% of student testers needed to know which cross-domain courses would be helpful to them in their future jobs. This is because they had no idea how to select courses crossing domains that were pertinent to their interests.
  • Most student testers agreed that matching courses with their interests is very important. Furthermore, they wanted to know the course details related to their future jobs. This evidence supports the study results presented in Section 2.
  • However, when facing a number of cross-domain courses, around 87% of students encountered difficulties in choosing the appropriate ones. A potential explanation is that they were in departments that required one knowledge domain.

Figure 7. Experimental results for the second insight on usage evaluation.

To provide the third insight, the questions were designed to assess the current course selection system. These evaluation results are shown in Figure 8, and the findings are highlighted below:

  • The results of the 1st and 2nd questions show that the current system cannot identify interests and cannot provide cross-domain recommendations. This reveals the shortcomings of the current system.
  • As a result, around 96% of student testers needed a service to recommend relevant cross-domain courses that were suitable for their future jobs.
  • Moreover, most of the student testers preferred being informed of their predicted scores for the recommended courses.
  • The above findings show that most of the student testers were not satisfied with the current system because of the lack of course recommendations.

Overall, Figures 6-8 support the practicality of the proposed idea because most students needed an intelligent system to facilitate cross-domain course selection. Finally, Figure 9 depicts the results for our proposed system, which can be summarized as follows. 

  • Over 90% of students provided positive ratings for the 1st and 3rd questions, which suggests that the proposed system can identify personal interests from the Holland code assessment and further recommend courses that will be helpful to the student’s future career.
  • Over 90% of students agreed that the system was able to predict course scores and that the predicted scores were helpful for ensuring more effective learning.
  • On the whole, around 95.54% of students were satisfied with the proposed system in terms of its recommendation and prediction.

Figure 8. Experimental results for the third insight on usage evaluation.

Figure 9. Experimental results of the fourth insight for usage evaluation.

  1. I suggest to further analyze more recent approaches about the examined topics. In particular, I suggest the following papers to investigate scientific community based approaches and emotional state in the introduction:

1) An emotional recommender system for music. IEEE Intelligent Systems.

2) Kira: a system for knowledge-based access to multimedia art collections. In 2017 IEEE 11th international conference on semantic computing (ICSC) (pp. 338-343).

Answer: Thanks for this comment. As suggested, we have added these related works [7, 8] into the references and cited them in Page 2.

  1. Finally, I suggest to perform a linguistic revision.

Answer: Thanks for this comment. Thanks for this comment. For this comment, we used the MDPI editing service to enhance the writing quality. Please refer to the following Certificate.

Round 2

Reviewer 3 Report

I think that the authors have addressed all my concerns